# Indoor running temporal variability for different running speeds, treadmill inclinations, and three different estimation strategies

**Andrea Zignoli**[1]*, **Antoine Godin**[2], **Laurent Mourot**[2]

**1** Department of Industrial Engineering, University of Trento, Trento, Italy, **2** Prognostic Factors and Regulatory Factors of Cardiac and Vascular Pathologies (EA3920), Exercise Performance Health Innovation (EPHI) platform, University of Franche-Comté, Besançon, France

* andrea.zignoli@unitn.it

## Abstract

Inertial measurement units (IMU) constitute a light and cost-effective alternative to gold-standard measurement systems in the assessment of running temporal variables. IMU data collected on 20 runners running at different speeds (80, 90, 100, 110 and 120% of preferred running speed) and treadmill inclination (±2, ±5, and ±8%) were used here to predict the following temporal variables: stride frequency, duty factor, and two indices of running variability such as the detrended fluctuation analysis alpha (DFA-α) and the Higuchi's D (HG-D). Three different estimation methodologies were compared: 1) a gold-standard optoelectronic device (which provided the reference values), 2) IMU placed on the runner's feet, 3) a single IMU on the runner's thorax used in conjunction with a machine learning algorithm with a short 2-second or a long 120-second window as input. A two-way ANOVA was used to test the presence of significant ($p<0.05$) differences due to the running condition or to the estimation methodology. The findings of this study suggest that using both IMU configurations for estimating stride frequency can be effective and comparable to the gold-standard. Additionally, the results indicate that the use of a single IMU on the thorax with a machine learning algorithm can lead to more accurate estimates of duty factor than the strategy of the IMU on the feet. However, caution should be exercised when using these techniques to measure running variability indices. Estimating DFA-α from a short 2-second time window was possible only in level running but not in downhill running and it could not accurately estimate HG-D across all running conditions. By taking a long 120-second window a machine learning algorithm could improve the accuracy in the estimation of DFA-α in all running conditions. By taking these factors into account, researchers and practitioners can make informed decisions about the use of IMU technology in measuring running biomechanics.

**Data Availability Statement:** Data samples will be uploaded at https://www.kaggle.com/datasets/andreazignoli/prissiv

**Funding:** This project was partially funded by ANTA with the ANTA Sports Award. The award has been presented at the 25th Annual Congress of the European College of Sport Science, in July 2020 in Seville, Spain. The prize has been awarded on the scientific merit of the proposal and to "stimulate research on next-generation sports science technology: uncovering insights in sports performance and injury prevention." The funders had no role in study design, data collection and analysis, decision to publish, or preparation of the manuscript.

**Competing interests:** The authors have declared that no competing interests exist.

## Introduction

Running temporal variables such as contact time, flight time, and duty cycle, are widely adopted in the study of the running biomechanics [1], and they are used to assess and characterize different running conditions [2]. Running structure variability [3], or simply 'variability' in the context of this manuscript, refers to a self-similarity assessment -computed on a stride-by-stride basis- of the running temporal variables. Two running variability indices of interest are the detrended fluctuation analysis long-range α (DFA-α) and the Higuchi's D (HG-D) [4]. Particularly, DFA-α is a measure of the correlation properties of a signal, and it is related to the size of fluctuations in signal changes with the length of time over which the fluctuations are measured. HG-D is a measure of the fractal dimension of a signal, which is a way of characterizing its overall complexity.

Variability is an intrinsic property of any temporal signal. However, a common practice in biomechanics is to assign to a locomotion pattern the characteristics of a signal with equal variability, e.g.: 1) DFA-α values closer to 1 (HG-D→1.8) are characteristics of the pink-noise, and hence of a flexible-and-adaptable locomotor behavior; 2) DFA-α values closer to 0.5 (HG-D→2) are characteristics of the white-noise, and hence of an impaired locomotion pattern which is not able to adapt to the environment; 3) DFA-α values closer to 0.75 (HG-D→1.9) indicate a locomotion pattern with both characteristics of structure and functional variance. These analogies have been widely adopted irrespectively from the locomotion strategy, e.g.: both in walking [5–8] and running [9–12].

Quite interestingly, Jordan et al. [9, 10] found a U-shaped relationship between running speed and DFA-α values computed with stride interval time. The minimum value of this relationship was found at the preferred running speed with a correspondent DFA-α value very close to 0.8. Decreasing variability has been observed after a heavy period of training [12] and in runners previously affected by a running-related injury [13]. The idea of monitoring variability to flag impaired running patterns potentially prone to injuries is therefore very appealing, but counts only few actual attempts [14–16].

To correctly assess running variability, accurate running temporal variables must be accurately estimated. Running variability is indeed best assessed in laboratory conditions, where optoelectronic devices, force platforms, motion capture systems, are available [17]. This instrumentation is too expensive and / or not suitable for outdoor conditions, where inertial measurement units (IMU) [18] or portable pressure insoles [19] are typically the best alternative. The basic assumption behind the use of IMU-solutions is that accelerations and rotational velocities collected at sensor level are reflective of the level ground running kinematics. Nowadays, IMUs are available off-the-shelf for sports applications and a smart watch, a smartphone or a laptop can be easily used to synchronize multiple IMUs placed on the runner's body. In the assessment of running kinematics, a single IMU can be placed close to the runner's center of mass (COM), therefore at the thorax, at the hip, or at the sacral level. Then, by adapting a spring-mass model of running [20], the estimated accelerations of the runner's COM can be used in the assessment of the kinematic variables [21]. A single IMU can minimize intrusiveness, but it can be poorly accurate in the estimation of kinematic variables. Another possible solution is to have multiple IMUs placed on the lower limbs, i.e.: at the foot or tibia level [22]. This configuration might provide more accurate estimations of the kinematic variables, but it might easily become distractive and invasive for the runner.

In parallel with the evolution of wearable technologies, machine learning algorithms (models) also found their way to a more accessible and easier implementation on running monitoring devices [23]. Machine learning models are especially useful in the estimation of running kinematic when sensory information is of limited accuracy or reliability, e.g.: in the case of

minimalist sensor configurations. Quite recently, Marotta et al. [24] adopted a random forest model to select the IMU configuration that could optimize intrusiveness and accuracy in running fatigue detection. They concluded that one to four IMUs (with at least one on the tibia) is required for running fatigue detection. Chow et al. [25] adopted a convolutional neural network (CNN) approach to evaluate running lower limb range of motion (knee and hip angles in the sagittal plane) with a single IMU sensor. Tan et al. [26] also used a CNN approach and a single IMU sensors placed either on the thorax, pelvis, thigh, shank, or foot: their goal was to establish the position of the IMU that could better estimate but their focus was on impact loading. Not surprisingly, the position which provided the best accuracy was the foot, followed by the pelvis, trunk, and thigh. Alcantara et al. [27] developed a recurrent neural network (RNN) to predict ground reaction forces from different variables, including: body mass, height, running speed, treadmill inclination, and percentage of a trial's steps classified as rearfoot, midfoot, or forefoot strikes.

In a preparatory technical work, the authors of this manuscript designed a machine learning model to predict running variability indices from a single IMU placed at the runner's thorax level [28]. The focus of the work was on the technical details and on the accuracy of the algorithm estimations. The present manuscript, which constitutes a natural expansion of the previous work, has two goals: 1) to present the values for indoor running variability at different speeds and treadmill inclinations collected with a gold-standard optoelectronic device, and 2) to present the comparison between three different possible estimation strategies for running kinematics and variability. The estimations strategies considered were: 1) the gold-standard optoelectronic device, 2) two IMUs placed on each foot of the runners, and 3) a single IMU placed on the runner's thorax used as input for a 2-second time window machine learning algorithm (already presented in [28]) or for a 120-second time window machine learning algorithm.

## Methods

### Data collection

Twenty recreational runners (11 males and 9 females; mean (SD); age 25 (6) years; BMI 20.76 (1.75) kg/m$^2$; running experience 5.9 (5.5) years; maximum aerobic speed 17.3 (1.8) km/h) volunteered for the study. Participants were recruited between January and June 2021. This study protocol was approved by the national ethical review board (2019- A03012-55) and was conducted in accordance with the guidelines of the Helsinki World Medical Association Declaration. Following the explanation of any risks and benefits associated with the study, all participants approved the study and gave written informed consent. Data/samples were collected between February and July 2021. The data post-process was conducted between July 2021 and July 2022. Only the authors who participated in data collection (i.e., AG and LM) had access to information that could identify individual participants during or after data collection.

The individual's self-selected preferred running speed (PRS) was assessed following the method described by [9, 13]. The participants were asked to run at 80, 90, 100, 110 and 120% of PRS, and to run at 90% of PRS at different treadmill inclinations: ±2, ±5, and ±8%, in a randomized order. Each running condition was held for 10 minutes. An IMU (Movesense, Suunto, Finland) was attached on the runner's thorax with a chest band, and two IMUs were fixed to each instep foot with a clip. An optoelectronic system (Optogait, Microgate, Italy) was used for the optical detection of the running kinematics. The Optogait transmitting and receiving bars were permanently fixed with adhesive-tape on the lateral bands of the treadmill.

## Temporal variables values (Optogait)

Gold-standard stride frequency and duty factor from the optoelectronic device were directly accessed after exporting the data with the Optogait software. The long-range DFA-α and HG-D were calculated from the stride frequency for all the 20 runners in all running conditions with the Python (3.8) package *antropy*. Both DFA-α and HG-D were computed considering the first 10 minutes after the start of the single running condition following best-practice guidelines [4]. Additional results including DFA and HG-D calculations at different times (2, 4, 6, 8) are available in the Additional Material https://drive.google.com/file/d/1y32xqVDioUXlzbsaLkgIL1JEVwg1kTXA/view?usp=sharingand in [28].

## Other estimation strategies (IMU on the feet)

In sync with the Optogait, foot-specific stride frequencies and duty factors were computed by averaging the stride frequencies and duty factors obtained with the IMUs fixed at the feet [22] in the same time window. After trial and error, authors decided to use the peaks of the norm of the accelerations to estimate both the stride frequency and the duty factor, which is one of the alternatives proposed in [22]. Both DFA-α and HG-D were computed as specified in the 'Temporal variables values' Section.

## Other estimation strategies (IMU on the thorax + machine learning algorithm with short time windows)

In sync with the Optogait, the IMU placed on the runner's thorax was used to provide the input for the machine learning algorithm (i.e., the input variables), whilst the optoelectronic device was used to provide the output (reference values that the neural network must learn to reproduce, i.e.: the output variables). Two datasets including the data from 10 runners each were created: the *training* dataset and the *test* dataset. The former was used to train the machine learning algorithm (i.e., to adjust the parameters of the model), whilst the latter was used to assess the accuracy of the model estimations. The technical details of the machine learning algorithm have been recently discussed and presented in peer-reviewed conference paper [28]. Here, they are briefly presented and summarized in **Fig 1**. A 2-sec rolling window was used to select an input with fixed size every second. The input consisted therefore in 6 IMU variables taken for 2-sec at a sampling frequency of 208 Hz, which resulted in a 6x416 data matrix. A new input matrix was generated every second. The 6 IMU variables consisted in the three linear accelerations along the three IMU axes ($a_x$, $a_y$, and $a_z$) and the three angular velocities about the IMU axes ($\omega_x$, $\omega_y$, and $\omega_z$).

For each input matrix, the acceleration values were normalized on 9.81 m/s$^2$ and the angular velocities were normalized by 500 rad/sec. The gold-standard foot-specific stride frequencies and duty factors were obtained by averaging the optoelectronic stride frequencies and duty factors provided in the same time window. Therefore, the output of the machine learning algorithm has size of 4x1. To train the algorithm, the values of the labels were z-standardized by subtracting the average value and normalizing by the standard deviation. The standardization procedure is commonly performed to improve the stability of the training process. In more detail, the machine learning model consisted in a CNN developed in Python (3.8) with the *Keras* library within the *Tensorflow* (2.7) framework. Once the estimated stride frequency was available, both DFA-α and HG-D were computed as specified in the 'Temporal variables values' Section. More details about the structure of the CNN can be found in [28]. All the *Python* scripts are available at the project repository (https://bitbucket.org/andrea_zignoli/prissiv/src/np/src/). Additionally, a Shiny web app has been developed in RStudio with the

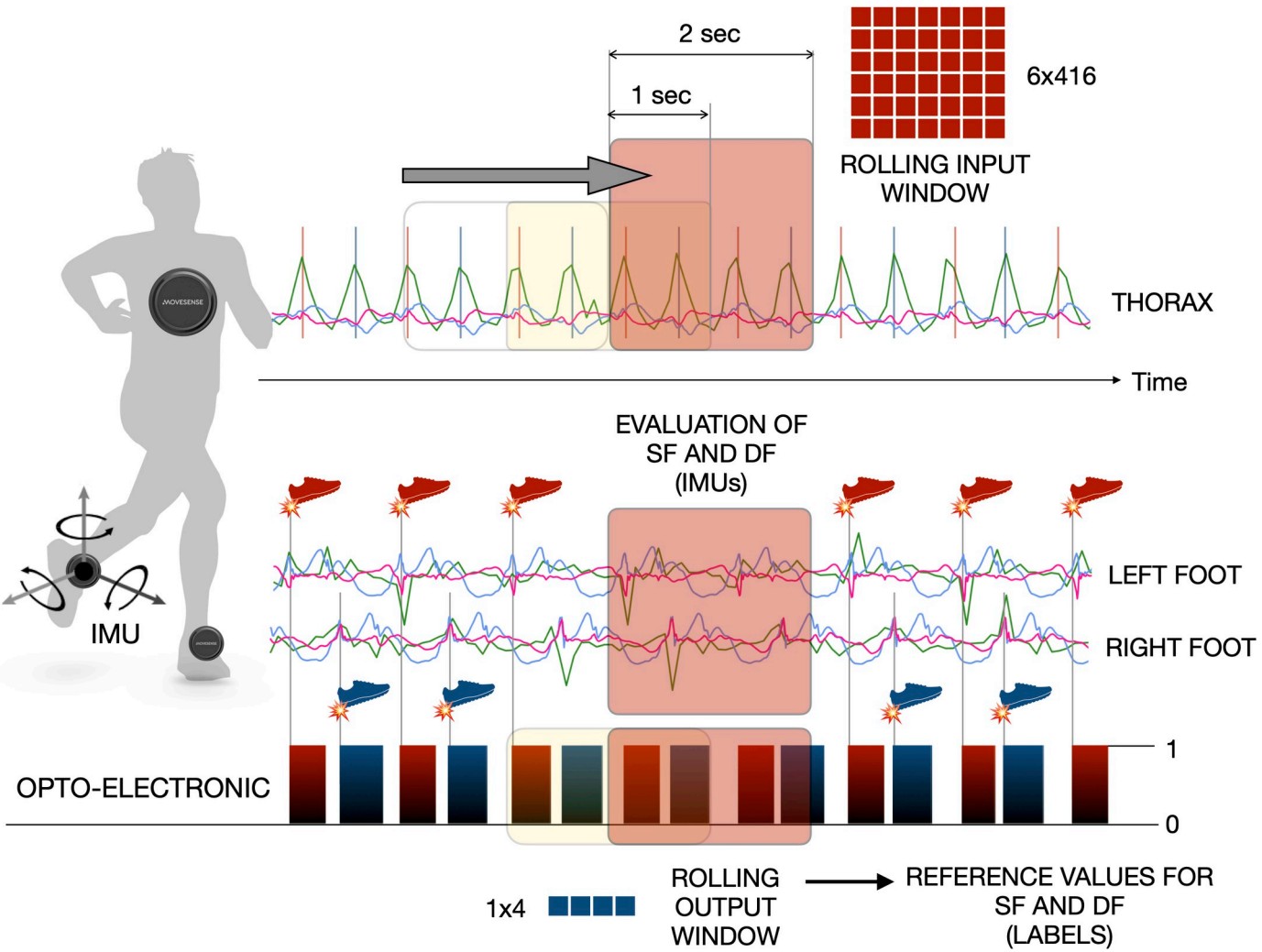

**Fig 1. Schematic representation of the data pipeline from collection to output calculation.** Runners were equipped with three IMU sensors placed at the thorax and foot levels. Every IMU sensors can collect at 208 Hz data from linear accelerations ($a_x$, $a_y$, and $a_z$) and rotation speeds ($\omega_x$, $\omega_y$, and $\omega_z$) along and about the three local IMU axes (x, y, and z). A rolling 2-sec window is displaced at 1-sec steps to derive the input data, which consist in a 6x416 tensor, from the IMU positioned on the runner's thorax. The IMUs placed at the foot level are used to derive an estimation of the stride frequency and the duty factor each foot. An optoelectronic device is also used to collect reference values for the stride frequency and duty factor. The data provided by the optoelectronic device are used as labels in the supervised training of the machine learning algorithm.

*shiny* R package. It is free to use and available for inference at andreazignoli.shinyapps.io/view_data. More details about the app and the app usage are available at the link of the project repository.

## Other estimation strategies (IMU on the thorax + machine learning algorithm with long time windows for DFA-α)

In the previous section, a machine learning model designed to estimate the duty factor and the stride frequency from short 2-sec time windows was introduced. With this methodology, DFA-α and HG-D were directly computed from the estimated stride frequency. However, while stride frequency can provide information about the number of strides per unit of time, might not be able to capture the complexity or structure of the signal. Variability indices, on

the other hand, evaluate the correlation properties within the signal, considering the relationship between different time scales. This is why a machine learning algorithm able to directly estimate DFA-α from a long-time window of 120 seconds of IMU data was developed.

This algorithm consists of a deep neural network developed in Python (3.8) with dense layers provided by the *Keras* library in the *Tensorflow* (2.7) framework. Three layers were included of 128, 64, and 32 neurons each, and the layers were interspersed with batch normalisation and dropout layers. The algorithm was trained to predict a single value of DFA-α from a 120-second window of IMU data. Considering that 120 seconds of data from an IMU collected at 208 Hz provides a large input matrix, a subset of temporal features were extracted from the linear accelerations $a_x$, $a_y$, $a_z$ and the angular velocities about the IMU axes $\omega_x$, $\omega_y$, and $\omega_z$. The selected temporal features for each temporal signal were as follows: skewness and Kurtosis indices (with *scipy* Python library), HG-D and DFA-α (with *antropy* Python library), standard deviation and average, spectral centroid, entropy and peak frequency, autocorrelation factors (with *scipy* Python library), and wavelet transform (with *pywavelets* Python library). These features were computed for each IMU signal, resulting in a 6 x 23 input matrix for the deep neural network. The input data was provided by the IMU positioned on the runner's thorax, and the DFA-α data used to train the model were provided by the gold-standard optoelectronic device (average between right and left foot). To train the model, the same procedure detailed in the previous Section was repeated, and 120-second window data were taken from the same individuals included in the training dataset with steps of 20 seconds. Accuracy was evaluated for every running condition on individuals not included in the training dataset.

## Statistical analysis

Statistical analysis was conducted with custom scripts written in R programming language. Specifically, the graphical user interface RStudio (ver. 2021.09.1) was used to write the scripts and the *ggstatsplot* R library [29] was used to conduct the analyses and plot the results.

First, running kinematic and variability data was organized into several groups based on one single-factor variable (i.e., the running condition: different running speeds and treadmill inclinations). One-way ANOVA was used to test whether the averages of the different groups were equal/not equal to the others. Additional multiple pairwise-comparisons were conducted if the average difference between specific pairs of groups were statistically significant. For this analysis, all the data (training and test dataset) was used. Second, the running temporal and variability data was also organized into several groups based on two single-factor variables (i.e., running condition and estimation strategy). A two-way ANOVA was used to test whether the averages of the different groups were equal/not equal to the others. Additional multiple pairwise-comparisons were conducted if the average difference between specific pairs of groups were statistically significant. For this latter analysis, only data belonging to the test dataset was used. Also, the root mean square error (RMSE) was computed between the temporal parameters estimated with the IMU-based strategies and the gold standard optoelectronic device.

Before conducting the analyses, a Shapiro-Wilk test was conducted on a group-by-group basis to check for normality. If the assumption of normality was confirmed/violated in the majority of the groups, a parametric/non-parametric version of the analysis was conducted. By default, the *ggstatsplot* R library [29] reports the F-Welch's statistics, the corresponding p-value, and the $\omega^2$ effect size (with confidence intervals [$CI_{95\%}$]) for the parametric version of the ANOVA test, and conducts a Games-Howell's post-hoc analysis with Holm's correction. Conversely, the library reports the $\chi^2$-Kruskal-Wallis' statistics, the corresponding p-value, and the $\varepsilon^2$ effect size (with [$CI_{95\%}$]) for the non-parametric version of the ANOVA test and

conducts a Dunn's post-hoc analysis with Holm's correction. For all the analyses, significance level was set at p = 0.05.

All the code used in this study to generate the statistics (R scripts) is available publicly in the project repository (https://bitbucket.org/andrea_zignoli/prissiv/src/np/src/).

## Results

An exhaustive and complete presentation of all the analyses is available in the S1 File.

### Temporal variables values

The average values together with the standard deviations and the results of the normality tests for all the temporal variables are reported in **Table 1**.

Normality test on running frequency returned a p>0.05 in every case except for +8 running condition (p = 0.007) (**Table 1**). Given these results, a parametric version of the ANOVA test was conducted on the running frequency. ANOVA conducted on running frequency with different running speed conditions returned p = 0.0026 ($\omega^2$ = 0.07), so running frequencies are different for different running speeds. Post-hoc analysis revealed significant differences in stride frequency between both 80 and 90 versus 120%PRS (p = 0.003 and p = 0.02, respectively). ANOVA conducted on running frequency with different treadmill inclinations returned p = 0.13 ($\omega^2$ = 0.02), suggesting that the running frequencies are not different across treadmill inclinations.

Normality test on duty factor returned a p>0.05 in every case except for few running conditions where significance was very closely approached (**Table 1**). Given these results, a parametric version of the ANOVA test was conducted on the duty factor. ANOVA conducted on duty factor with different running speed conditions returned p<0.001 ($\omega^2$ = 0.48), so this means that duty factor is speed dependent. Post-hoc analysis revealed significant differences in duty factor for 80%PRS vs all the other speeds [80 vs 90 (p<0.001), 80 vs 100 (p<0.001), 80 vs 110 (p<0.001), 80 vs 120 (p<0.001)]. ANOVA conducted on duty factor with different treadmill inclinations returned p = 0.14 ($\omega^2$ = 0.02), so the hypothesis that all average duty factors are different for different treadmill inclinations is rejected.

Normality test on DFA-α returned a p>0.05 in every case except for some running conditions where significance was not closely approached (**Table 1**). Given these results, a non-parametric version of the ANOVA test was conducted on DFA-α. ANOVA conducted on DFA-α with different running speeds returned p = 0.07 ($\omega^2$ = 0.05), so the hypothesis that all average DFA-α are different for different running speeds is rejected. However, a significant difference was found between running conditions 80%PRS versus 120%PRS (p = 0.04). ANOVA conducted on DFA-α with different treadmill inclinations returned p<0.001 ($\omega^2$ = 0.17), so the hypothesis that all average DFA-α are equal for different treadmill inclinations is rejected. Post-hoc analysis revealed significant differences in DFA-α for different treadmill inclinations: -8 vs +2 (p = 0.00237) and +5 (p = 0.00106) and +8 (p = 0.00749), -5 vs +2 (p = 0.0087) and +5 (p = 0.00467) and +8 (p = 0.02), -2 vs +2 (p = 0.02) and +5 (p = 0.01).

Normality test on HG-D returned a p>0.05 in every case except for some running conditions where significance was not closely approached (**Table 1**). Given these results, a non-parametric version of the ANOVA test was conducted on HG-D. ANOVA conducted on HG-D with different running speed conditions returned p = 0.24 ($\omega^2$ = 0.03), so the hypothesis that all average HG-D are different for different running speeds is rejected. ANOVA conducted on HG-D with different treadmill inclinations returned p = 0.18 ($\omega^2$ = 0.04), so the hypothesis that all average HG-D are different for different treadmill inclinations is rejected.

**Table 1. Values (average (Avg.) and standard deviation (Sd.)) and results of the normality test analysis (p-values) for stride frequency (SF), duty factor (DF), and running variability DFA-α and HG-D (HG-D)) for different running speeds and treadmill inclinations.** Values are reported for the three estimation strategies: the gold-standard optoelectronic device (OPTO), the IMU sensors attached to the runners' feet (MV), and the single IMU sensor attached to the runner's thorax used as input for the machine learning algorithm (NN). For OPTO, all the values refer to the entire database (20 runners), whilst for MV and NN they refer to the validation dataset (10 runners). Additionally, NN-long-refers to a machine learning algorithm that can process IMU features from a long 120-second window and estimate DFA-α.

| Running cond. | OPTO-p | OPTO-Avg. | OPTO-Sd. | MV-p | MV-Avg. | MV-Sd. | NN-p | NN-Avg. | NN-Sd. | Vars. | NN-long-p | NN-long-Avg. | NN-long-Sd. | Unit |
|---|---|---|---|---|---|---|---|---|---|---|---|---|---|---|
| -8% | 0.147 | 1.355 | 0.069 | 0.010 | 1.394 | 0.052 | 0.032 | 1.396 | 0.026 | SF | | | | Hz |
| -5% | 0.293 | 1.368 | 0.075 | 0.002 | 1.399 | 0.051 | 0.048 | 1.389 | 0.037 | SF | | | | Hz |
| -2% | 0.389 | 1.386 | 0.081 | 0.229 | 1.403 | 0.069 | 0.257 | 1.387 | 0.048 | SF | | | | Hz |
| 2% | 0.149 | 1.384 | 0.082 | 0.013 | 1.408 | 0.058 | 0.024 | 1.391 | 0.040 | SF | | | | Hz |
| 5% | 0.111 | 1.393 | 0.075 | 0.006 | 1.412 | 0.056 | 0.143 | 1.392 | 0.039 | SF | | | | Hz |
| 8% | 0.007 | 1.405 | 0.087 | 0.001 | 1.420 | 0.069 | 0.005 | 1.393 | 0.049 | SF | | | | Hz |
| 80%PRS | 0.171 | 1.359 | 0.072 | 0.065 | 1.386 | 0.060 | 0.026 | 1.387 | 0.036 | SF | | | | Hz |
| 90%PRS | 0.443 | 1.369 | 0.080 | 0.251 | 1.391 | 0.064 | 0.223 | 1.389 | 0.045 | SF | | | | Hz |
| 100%PRS | 0.306 | 1.386 | 0.087 | 0.356 | 1.410 | 0.065 | 0.427 | 1.385 | 0.049 | SF | | | | Hz |
| 110%PRS | 0.149 | 1.405 | 0.090 | 0.189 | 1.438 | 0.079 | 0.429 | 1.397 | 0.056 | SF | | | | Hz |
| 120%PRS | 0.144 | 1.435 | 0.096 | 0.118 | 1.463 | 0.081 | 0.377 | 1.406 | 0.058 | SF | | | | Hz |
| -8% | 0.041 | 41.151 | 2.707 | 0.257 | 0.510 | 0.042 | 0.043 | 0.401 | 0.021 | DF | | | | % |
| -5% | 0.341 | 41.424 | 3.316 | 0.514 | 0.496 | 0.027 | 0.090 | 0.405 | 0.018 | DF | | | | % |
| -2% | 0.355 | 42.338 | 2.656 | 0.002 | 0.488 | 0.021 | 0.025 | 0.392 | 0.035 | DF | | | | % |
| 2% | 0.238 | 42.637 | 2.915 | 0.198 | 0.496 | 0.034 | 0.001 | 0.399 | 0.031 | DF | | | | % |
| 5% | 0.060 | 42.783 | 3.198 | 0.046 | 0.490 | 0.033 | 0.024 | 0.401 | 0.034 | DF | | | | % |
| 8% | 0.146 | 42.277 | 2.713 | 0.003 | 0.485 | 0.018 | 0.012 | 0.392 | 0.032 | DF | | | | % |
| 80%PRS | 0.380 | 44.776 | 3.168 | 0.000 | 0.498 | 0.025 | 0.004 | 0.405 | 0.032 | DF | | | | % |
| 90%PRS | 0.047 | 41.698 | 3.026 | 0.043 | 0.493 | 0.036 | 0.026 | 0.390 | 0.037 | DF | | | | % |
| 100%PRS | 0.055 | 40.121 | 2.826 | 0.020 | 0.494 | 0.049 | 0.346 | 0.381 | 0.035 | DF | | | | % |
| 110%PRS | 0.042 | 38.104 | 2.958 | 0.003 | 0.494 | 0.062 | 0.007 | 0.381 | 0.042 | DF | | | | % |
| 120%PRS | 0.036 | 36.554 | 2.831 | 0.001 | 0.507 | 0.062 | 0.061 | 0.375 | 0.041 | DF | | | | % |
| -8% | 0.194 | 0.568 | 0.153 | 0.091 | 0.553 | 0.176 | 0.068 | 0.800 | 0.076 | DFA-α | 0.624 | 0.50 | 0.07 | - |
| -5% | 0.524 | 0.571 | 0.191 | 0.006 | 0.592 | 0.155 | 0.471 | 0.752 | 0.081 | DFA-α | 0.700 | 0.50 | 0.06 | - |
| -2% | 0.062 | 0.583 | 0.200 | 0.123 | 0.562 | 0.143 | 0.008 | 0.836 | 0.071 | DFA-α | 0.132 | 0.52 | 0.09 | - |
| 2% | 0.633 | 0.715 | 0.135 | 0.008 | 0.563 | 0.196 | 0.246 | 0.814 | 0.082 | DFA-α | 0.702 | 0.58 | 0.08 | - |
| 5% | 0.216 | 0.715 | 0.068 | 0.004 | 0.525 | 0.164 | 0.313 | 0.798 | 0.051 | DFA-α | 0.907 | 0.59 | 0.07 | - |
| 8% | 0.003 | 0.729 | 0.157 | 0.025 | 0.514 | 0.159 | 0.033 | 0.796 | 0.047 | DFA-α | 0.349 | 0.58 | 0.04 | - |
| 80%PRS | 0.919 | 0.691 | 0.103 | 0.005 | 0.563 | 0.188 | 0.007 | 0.788 | 0.064 | DFA-α | 0.255 | 0.57 | 0.08 | - |
| 90%PRS | 0.146 | 0.733 | 0.075 | 0.179 | 0.566 | 0.154 | 0.168 | 0.790 | 0.086 | DFA-α | 0.720 | 0.60 | 0.07 | - |
| 100%PRS | 0.635 | 0.734 | 0.106 | 0.016 | 0.519 | 0.154 | 0.100 | 0.791 | 0.071 | DFA-α | 0.333 | 0.60 | 0.08 | - |
| 110%PRS | 0.097 | 0.755 | 0.134 | 0.014 | 0.535 | 0.183 | 0.662 | 0.800 | 0.076 | DFA-α | 0.924 | 0.58 | 0.07 | - |
| 120%PRS | 0.000 | 0.801 | 0.180 | 0.088 | 0.538 | 0.151 | 0.320 | 0.793 | 0.039 | DFA-α | 0.950 | 0.58 | 0.07 | - |
| -8% | 0.062 | 2.001 | 0.029 | 0.340 | 2.010 | 0.037 | 0.003 | 1.912 | 0.032 | HG-D | | | | - |
| -5% | 0.912 | 2.003 | 0.029 | 0.008 | 2.003 | 0.033 | 0.182 | 1.925 | 0.030 | HG-D | | | | - |

(*Continued*)

**Table 1.** (Continued)

| Running cond. | OPTO-p | OPTO-Avg. | OPTO-Sd. | MV-p | MV-Avg. | MV-Sd. | NN-p | NN-Avg. | NN-Sd. | Vars. | NN-long-p | NN-long-Avg. | NN-long-Sd. | Unit |
|---|---|---|---|---|---|---|---|---|---|---|---|---|---|---|
| -2% | 0.291 | 2.009 | 0.033 | 0.034 | 2.003 | 0.037 | 0.004 | 1.897 | 0.035 | HG-D | | | | - |
| 2% | 0.404 | 1.997 | 0.027 | 0.199 | 1.995 | 0.042 | 0.099 | 1.914 | 0.030 | HG-D | | | | - |
| 5% | 0.237 | 1.993 | 0.030 | 0.241 | 2.017 | 0.036 | 0.132 | 1.932 | 0.015 | HG-D | | | | - |
| 8% | 0.226 | 1.995 | 0.026 | 0.102 | 2.027 | 0.028 | 0.010 | 1.936 | 0.022 | HG-D | | | | - |
| 80%PRS | 0.018 | 1.994 | 0.033 | 0.069 | 2.014 | 0.034 | 0.162 | 1.917 | 0.026 | HG-D | | | | - |
| 90%PRS | 0.046 | 1.984 | 0.031 | 0.103 | 2.011 | 0.034 | 0.205 | 1.908 | 0.029 | HG-D | | | | - |
| 100%PRS | 0.300 | 1.985 | 0.035 | 0.067 | 2.011 | 0.029 | 0.078 | 1.921 | 0.025 | HG-D | | | | - |
| 110%PRS | 0.002 | 1.983 | 0.042 | 0.001 | 2.005 | 0.046 | 0.127 | 1.914 | 0.024 | HG-D | | | | - |
| 120%PRS | 0.286 | 1.976 | 0.037 | 0.296 | 2.009 | 0.030 | 0.205 | 1.922 | 0.027 | HG-D | | | | - |

## Other estimation strategies

The values of the average differences in terms of RMSE are reported in **Table 2**.

A parametric version of the two-way ANOVA was conducted on the stride frequency. This returned a p>0.05 for all the different running speeds and treadmill inclinations, so the hypothesis that all average stride frequencies are different for different estimation strategies is rejected. The p-value for the interaction between running condition and estimation strategy was 0.9052 (across different speeds) and 0.9998 (among different treadmill inclinations), which indicates that there is no interaction between running condition and estimation strategy. Values and results of the statistics are reported in **Figs 2 and 3**.

A parametric version of the two-way ANOVA was conducted on the duty factor. This returned p = 0.037 for the speed factor and p<0.0001 for the estimation strategy. This is due to the significant differences between the estimations with the IMU on the feet and the two other estimations strategies. The p-value for the interaction between running condition and estimation strategy is 0.2123, which indicates that there is no interaction between running speed and estimation strategy. Two-way ANOVA conducted on duty factor with different treadmill inclinations returned p = 0.5511 for the treadmill inclination factor and p<0.0001 for the estimation

**Table 2. Root mean square errors of the temporal variables (stride frequency SF, duty factor (DF), DFA-α and Higuchi's D (HG-D)) for the two IMU-based estimation strategies (i.e.: IMU attached at the runners' feet (MV) and IMU attached at the runners' thorax + machine learning algorithm (NN)) with reference to the gold-standard optoelectronic device (OPTO) (in bold if significantly different from gold standard).** Additionally, DFA-α NN-LONG-OPTO includes the root mean square error between the machine learning algorithm that include IMU features from a long 120-second window and OPTO.

| Running cond. | SF MV-OPTO (Hz) | DF MV-OPTO (%) | DFA-α MV-OPTO (-) | HG-D MV-OPTO (-) | SF NN-OPTO (Hz) | DF NN-OPTO (%) | DFA-α NN-OPTO (-) | HG-D NN-OPTO (-) | DFA-α NN-LONG-OPTO (-) |
|---|---|---|---|---|---|---|---|---|---|
| -8 | 0.002 | **1.092** | 0.172 | 0.030 | 0.003 | 0.190 | **0.268** | **0.115** | 0.042 |
| -5 | 0.009 | 2.467 | 0.168 | 0.030 | 0.010 | 0.463 | 0.169 | **0.079** | 0.053 |
| -2 | 0.001 | **0.577** | 0.165 | 0.038 | 0.002 | 0.198 | **0.216** | **0.114** | 0.058 |
| 2 | 0.001 | **0.610** | 0.199 | 0.032 | 0.001 | 0.224 | 0.094 | **0.093** | 0.042 |
| 5 | 0.001 | **0.584** | 0.201 | 0.036 | 0.001 | 0.210 | 0.043 | **0.075** | 0.058 |
| 8 | 0.001 | **0.850** | 0.149 | 0.018 | 0.001 | 0.210 | 0.059 | **0.075** | 0.068 |
| 80 | 0.001 | **0.514** | 0.199 | 0.034 | 0.002 | **0.240** | 0.093 | **0.091** | 0.046 |
| 90 | 0.001 | **0.727** | 0.145 | 0.031 | 0.002 | 0.220 | 0.065 | **0.080** | 0.062 |
| 100 | 0.001 | **1.041** | 0.167 | 0.020 | 0.001 | 0.174 | 0.068 | **0.070** | 0.031 |
| 110 | 0.001 | **1.599** | 0.209 | 0.023 | 0.002 | 0.138 | 0.138 | **0.082** | 0.036 |
| 120 | 0.002 | **2.279** | 0.181 | 0.023 | 0.003 | 0.140 | 0.100 | **0.073** | 0.034 |

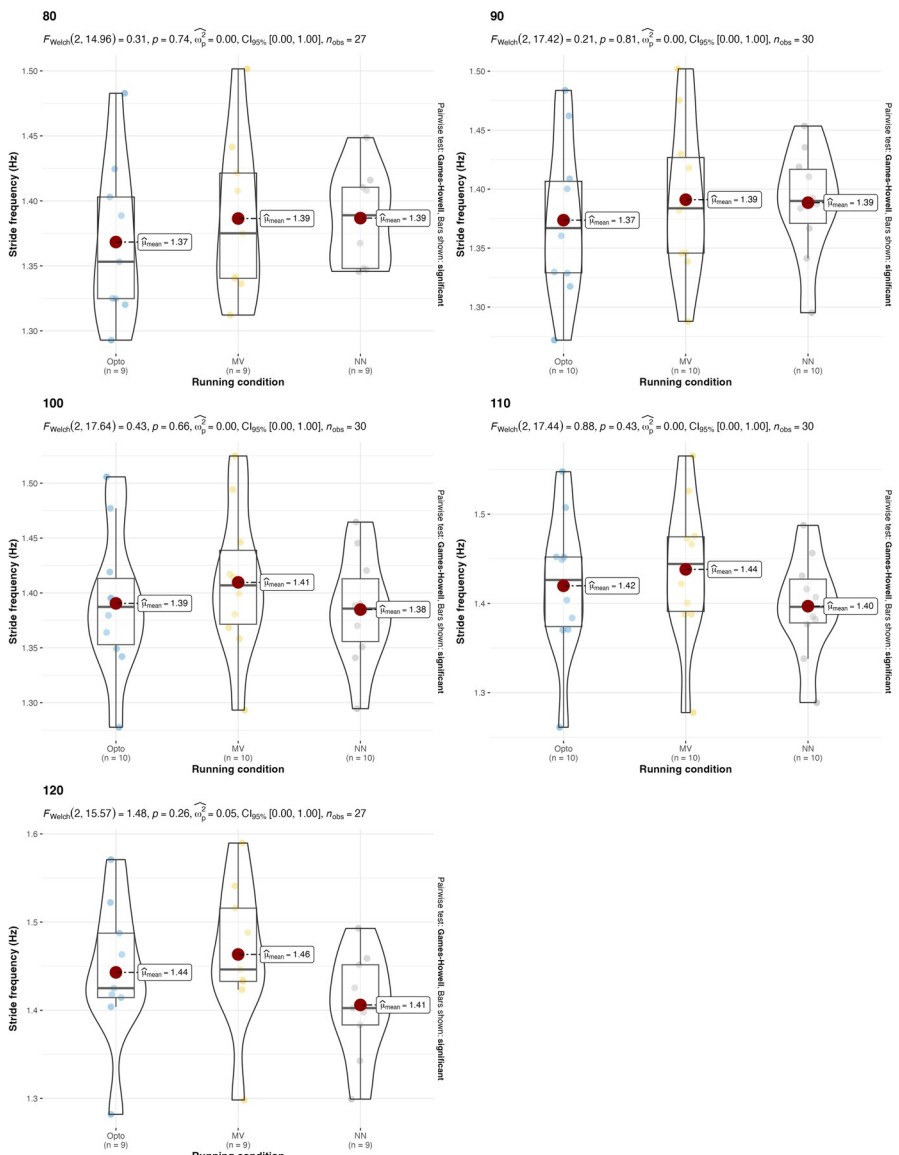

**Fig 2. Values and results of the statistical analysis for the stride frequency in different running speeds.** Values are reported for the three estimation strategies: the gold-standard optoelectronic device (OPTO), the IMU sensors attached to the runners' feet (MV), and the single IMU sensor attached to the runner's thorax used as input for the machine learning algorithm (NN).

strategy, so the hypothesis that all duty factor values are equal for different estimation strategies is rejected. The p-value for the interaction between running condition and estimation strategy is 0.5906, which indicates that there is no interaction between treadmill inclination and estimation strategy. Values and results of the statistics are reported in **Figs 4 and 5**.

A non-parametric version of the two-way ANOVA was conducted on DFA-α. Two-way ANOVA conducted on DFA-α with different running speeds returned p = 0.9333 for the speed factor and p<0.0001 for the estimation strategy, so the hypothesis that all DFA-α are equal for different estimation strategies is rejected. This is due to the significant differences between the estimations with the IMU on the thorax and the two other estimations strategies. The p-value for the interaction between running condition and estimation strategy is 0.9969,

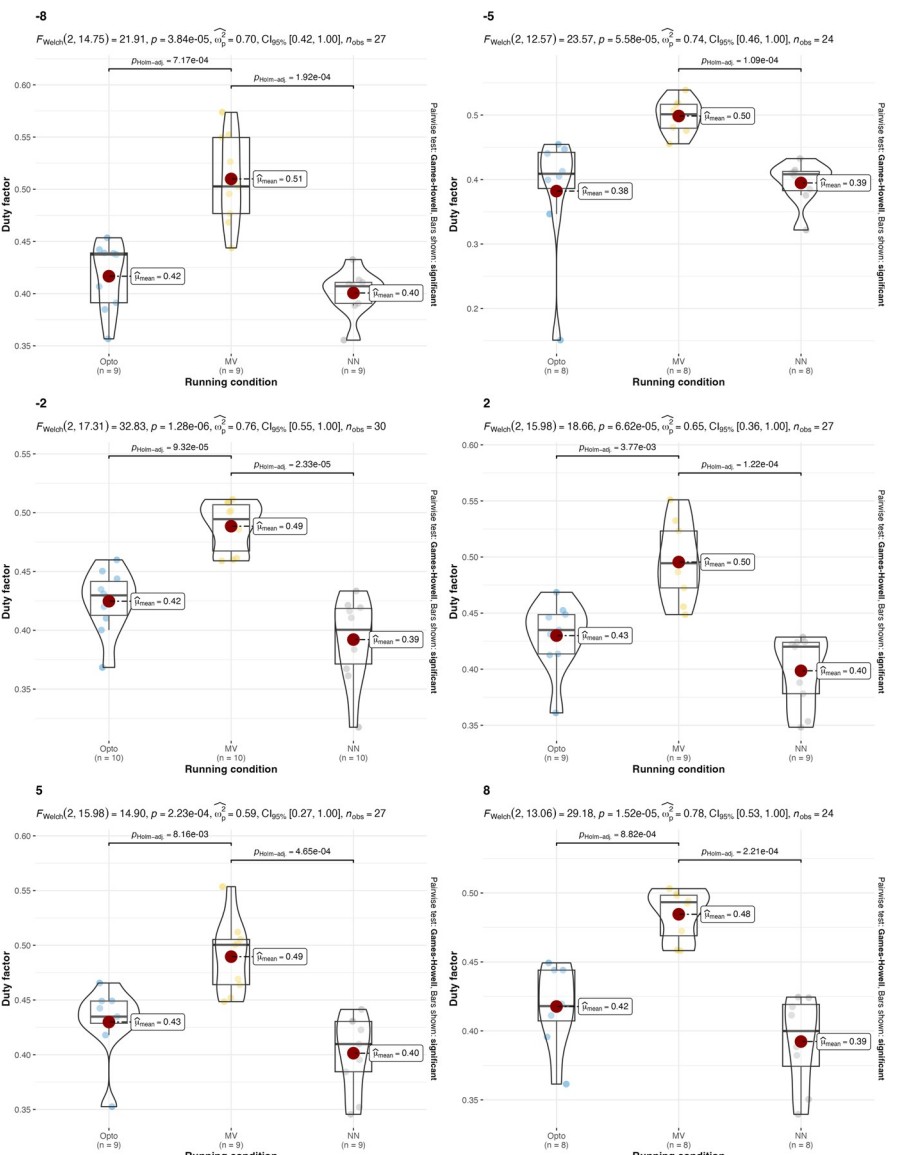

**Fig 3. Values and results of the statistical analysis for the stride frequency in different treadmill inclinations.**
Values are reported for the three estimation strategies: the gold-standard optoelectronic device (OPTO), the IMU sensors attached to the runners' feet (MV), and the single IMU sensor attached to the runner's thorax used as input for the machine learning algorithm (NN).

which indicates that there is no interaction between estimation strategy factor and running speed factor. Two-way ANOVA conducted on DFA-α with different treadmill inclinations returned p = 0.1759 for the treadmill inclination factor and p<0.0001 for the estimation strategy factor, so the hypothesis that all DFA-α are equal for different estimation strategies is rejected. The p-value for the interaction between running condition and estimation strategy is 0.0836, which indicates that there is no interaction between estimation strategy factor and treadmill inclination factor. Values and results of the statistics are reported in **Figs 6 and 7**.

A non-parametric version of the two-way ANOVA was conducted on HG-D. Two-way ANOVA conducted on HG-D with different treadmill inclinations returned p = 0.364 for the treadmill inclination factor and p<0.0001 for the estimation strategy, so the hypothesis that all

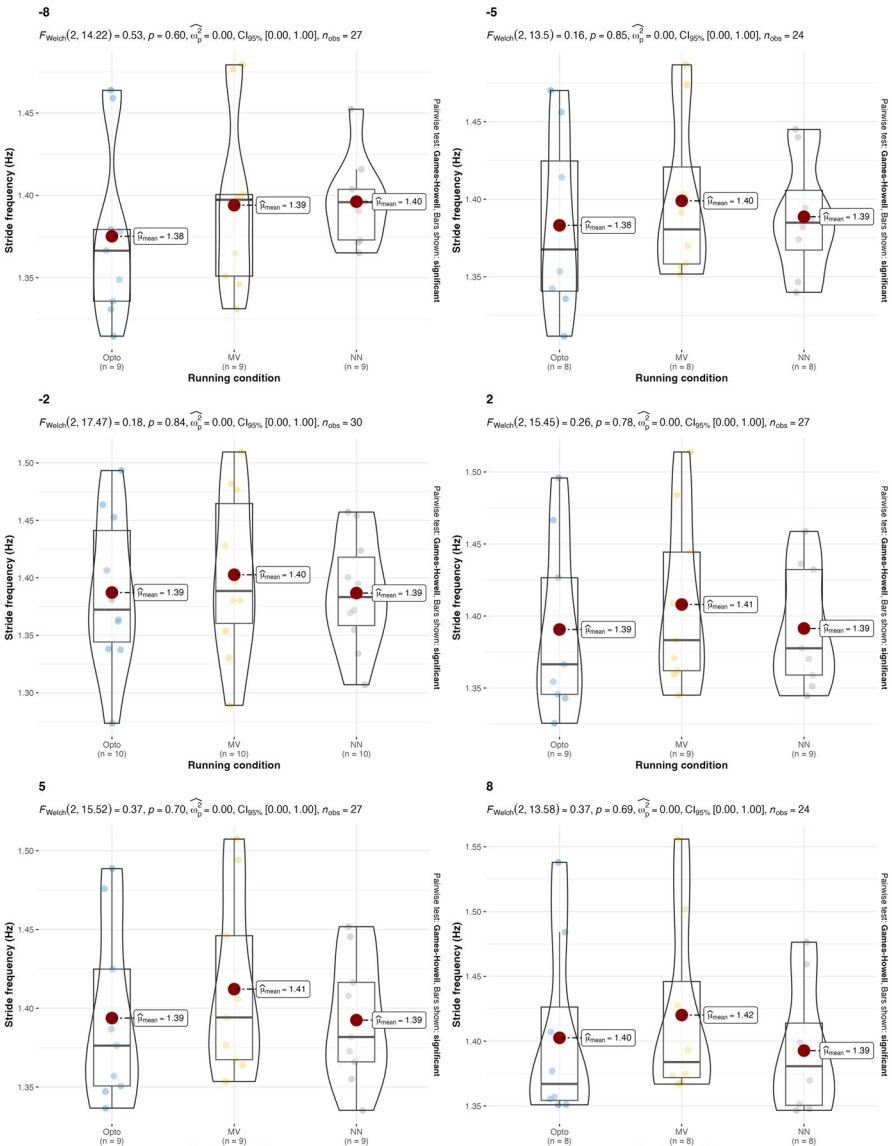

**Fig 4. Values and results of the statistical analysis for the duty factor in different running speeds.** Values are reported for the three estimation strategies: the gold-standard optoelectronic device (OPTO), the IMU sensors attached to the runners' feet (MV), and the single IMU sensor attached to the runner's thorax used as input for the machine learning algorithm (NN).

HG-D values are equal for different estimation strategies is rejected. The p-value for the interaction between running condition and estimation strategy is 0.6336, which indicates that there is no interaction between treadmill inclination and estimation strategy. Two-way ANOVA conducted on HG-D with different running speeds returned p = 0.6589 for the speed factor and p<0.0001 for the estimation strategy, so the hypothesis that all HG-D are equal for different estimation strategies is rejected. The p-value for the interaction between running condition and estimation strategy is 0.9899, which indicates that there is no interaction between estimation strategy factor and speed factor. Post-hoc analysis revealed that significant differences were present between different estimation strategies. Values and results of the statistics are reported in **Figs 8 and 9**.

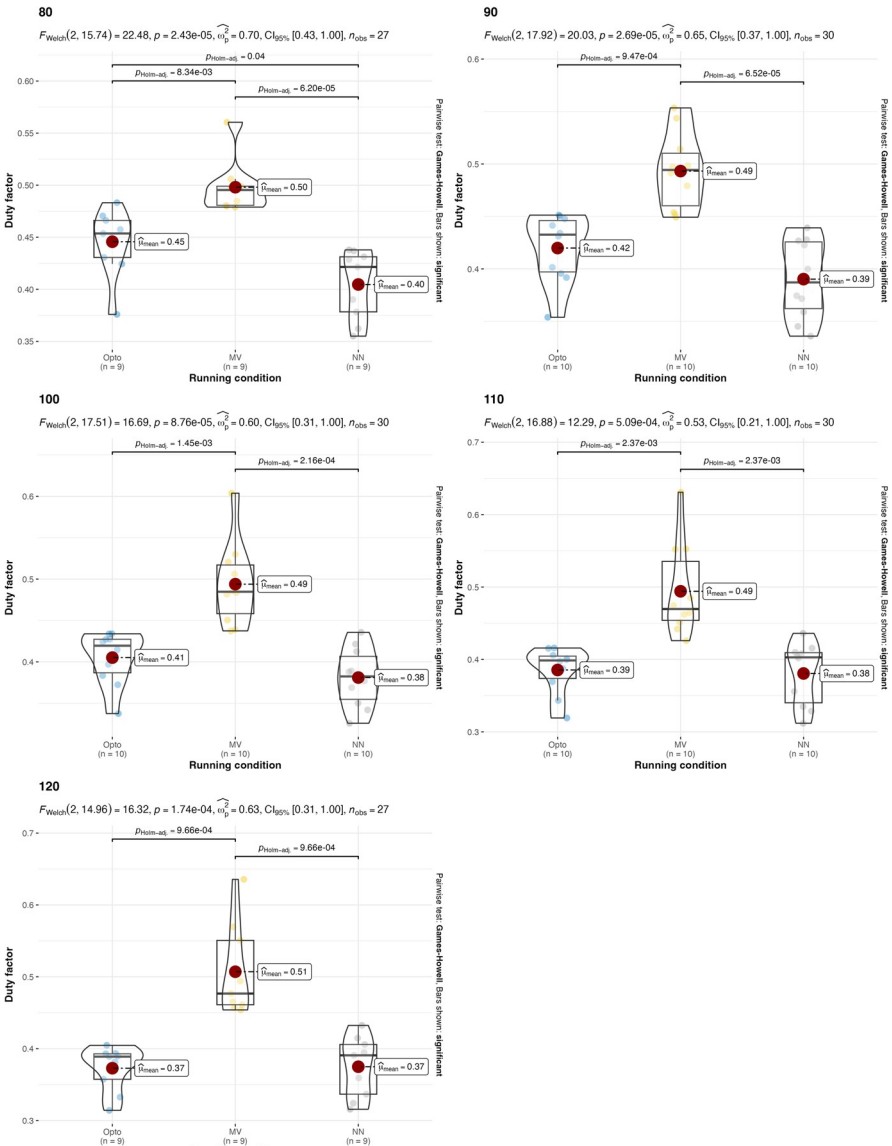

**Fig 5. Values and results of the statistical analysis for the duty factor in different treadmill inclinations.** Values are reported for the three estimation strategies: the gold-standard optoelectronic device (OPTO), the IMU sensors attached to the runners' feet (MV), and the single IMU sensor attached to the runner's thorax used as input for the machine learning algorithm (NN).

## Discussion

One of the goals of this work was to present values for indoor running kinematics (stride frequency and duty factor) and variability (DFA-α and HG-D) at different speeds and treadmill inclinations. A second goal of the work was to compare three different possible estimation strategies for running temporal variables: 1) a gold-standard optoelectronic device (Optogait), 2) a configuration where two IMUs were placed on each foot of the runners, and 3) a configuration where a machine learning algorithm was used to process the data collected with a single IMU placed on the runner's thorax (one version had the input from a short 2-second time window and another had the input from a long 120-second window). It has to be highlighted that

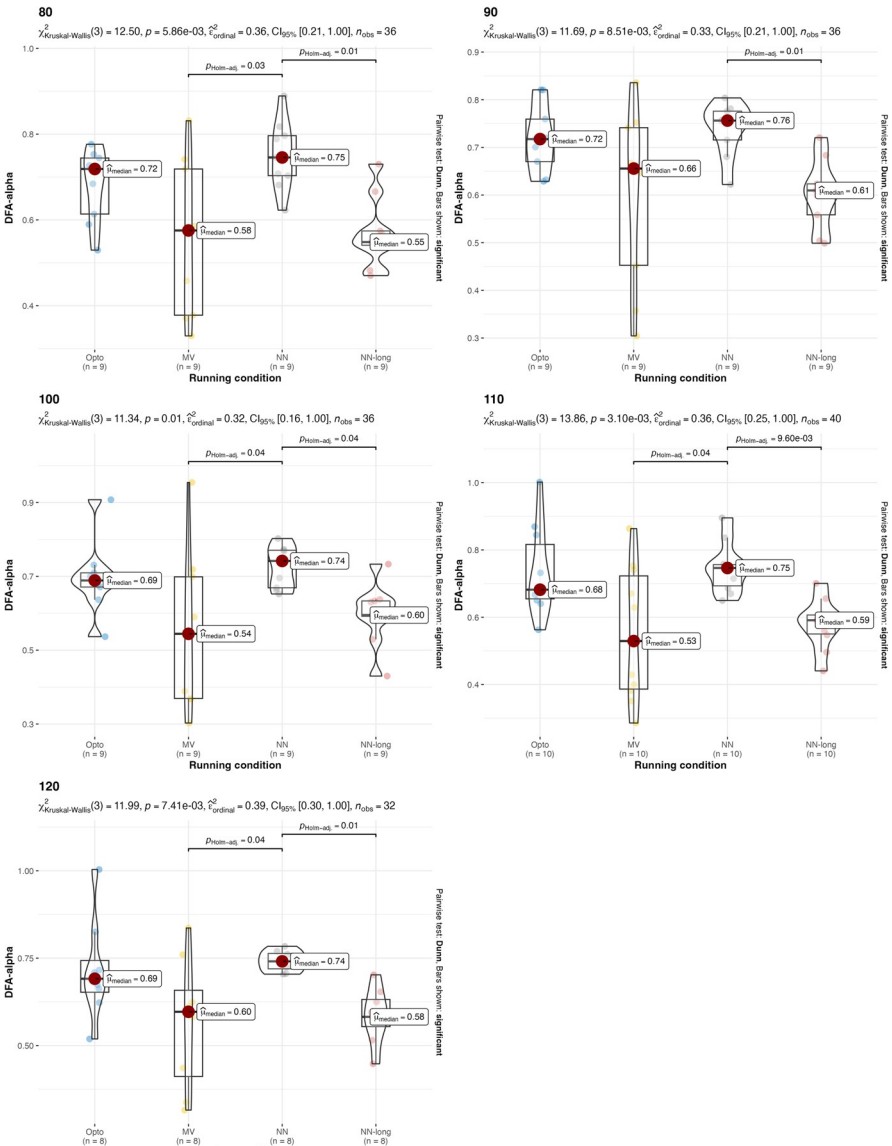

**Fig 6. Values and results of the statistical analysis for the running variability (DFA-α) in different running speeds.** Values are reported for the three estimation strategies: the gold-standard optoelectronic device (OPTO), the IMU sensors attached to the runners' feet (MV), and the single IMU sensor attached to the runner's thorax used as input for the machine learning algorithm (NN).

in this study a configuration where two IMUs were placed on each foot of the runner was not previously validated, so the results reported in this study might not be true for all the systems exploiting this sensor configuration. The results suggest that using both IMU-based configurations is effective for estimating stride frequency with good accuracy. Using a single IMU on the thorax together with a machine learning algorithm resulted in more accurate estimates of duty factor than the IMU attached to the runners' feet. Conversely, much more caution is advised when estimating running variability indices with IMU devices, especially in extreme downhill conditions.

In terms of stride frequency and duty factor, results are in line with what was previously summarized in authoritative reviews [2]: A) uphill running, compared to level running, is

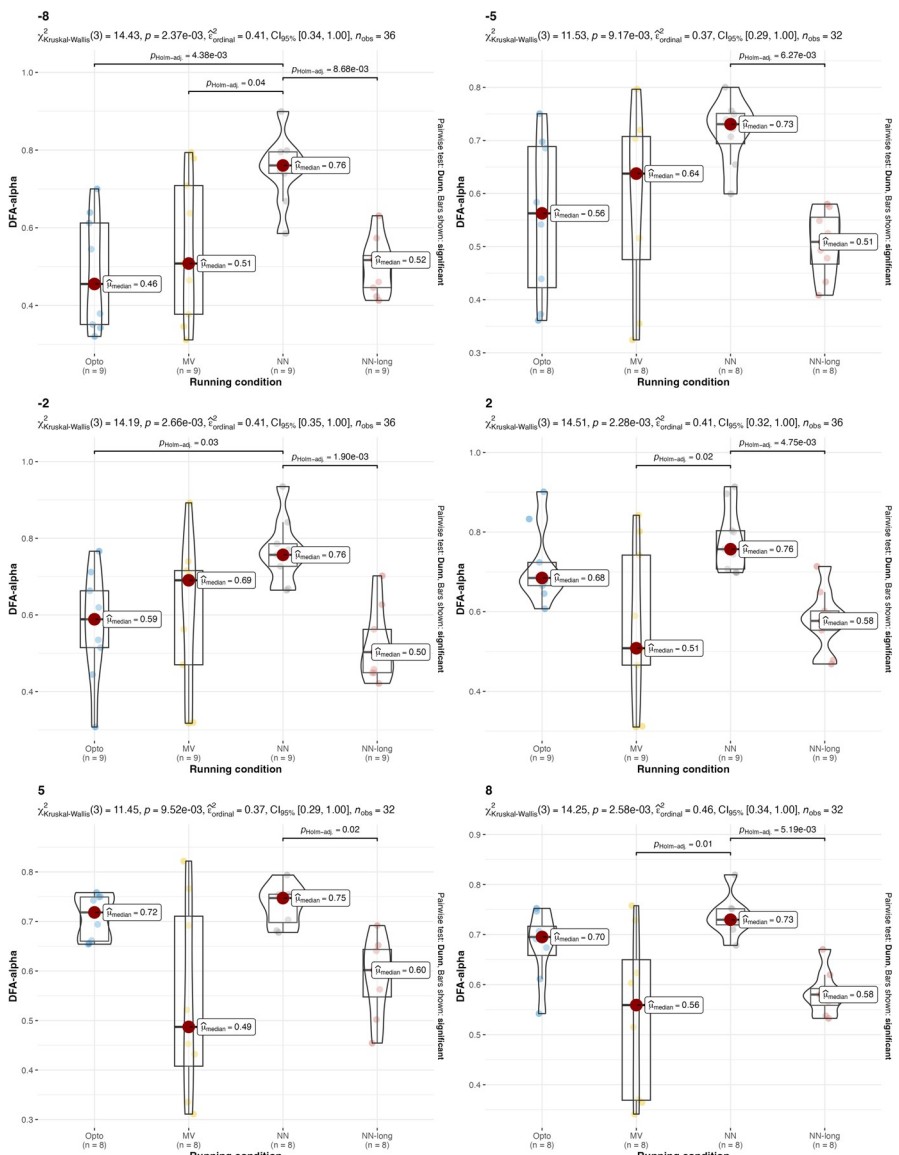

**Fig 7. Values and results of the statistical analysis for the running variability (DFA-α) in different treadmill inclinations.** Values are reported for the three estimation strategies: the gold-standard optoelectronic device (OPTO), the IMU sensors attached to the runners' feet (MV), and the single IMU sensor attached to the runner's thorax used as input for the machine learning algorithm with short 2-sec time window (NN) and 120-second long time window (NN-long).

typically characterized by a higher stride frequency and a higher duty factor; B) downhill running, compared to level running, is characterized by a reduced stride frequency and a similar duty factor. These results are consistent across all of the estimation strategies (**Table 1**).

In terms of variability indices, results suggest that no significant relationship was found between running speed and DFA-α or HG-D. The values found for DFA-α or HG-D at PRS did not differ for any of the other speeds. A single significant difference was found between DFA-α at 80 vs 120%PRS and this might indicate a possible trend but fitting the data with a quadratic (U-shaped) relationship [10] was not justifiable. Significant differences were instead found between DFA-α at different treadmill inclinations: DFA-α was found to be significantly

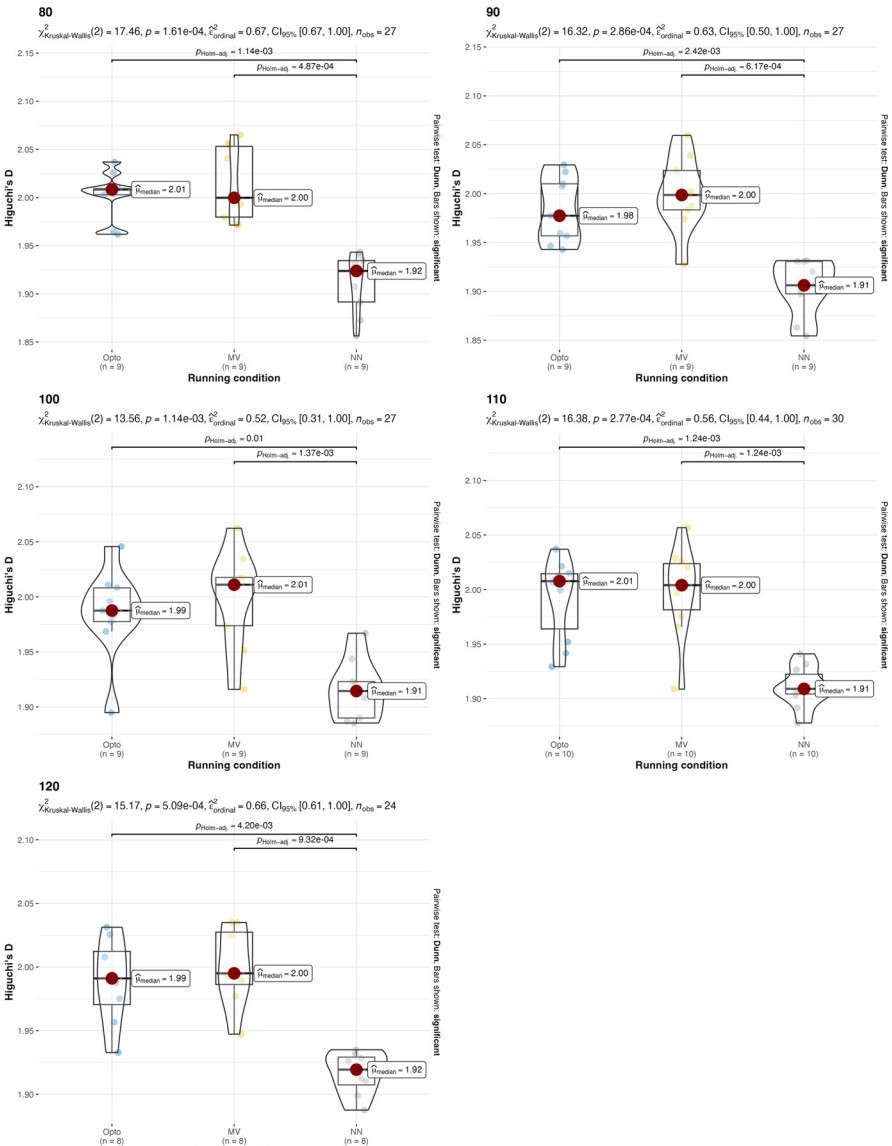

**Fig 8. Values and results of the statistical analysis for the running variability (HG-D) in different running speeds.**
Values are reported for the three estimation strategies: the gold-standard optoelectronic device (OPTO), the IMU
sensors attached to the runners' feet (MV), and the single IMU sensor attached to the runner's thorax used as input for
the machine learning algorithm with short 2-sec time window (NN) and 120-second long time window (NN-long).

lower in downhill vs uphill running (**Table 1**). A similar, although not significant, trend was
found for HG-D values: HG-D has the tendency to be lower in uphill vs downhill running.
These results suggest that uphill treadmill running is characterized by stronger long-term rela-
tionships than downhill treadmill running. Consequently, uphill treadmill running looks more
flexible and adaptable than downhill treadmill running.

One limitation of using IMUs to assess running kinematics and variability is the physical
distance between the point where the reaction force is generated (i.e., the ground-foot inter-
face) and the point where the accelerations are measured (e.g., foot instep, tibia, wrist, hip, tho-
rax). As a result, the kinematic data provided by an IMU only reflect the level ground
movements and do not capture the full extent of the movements and forces generated during

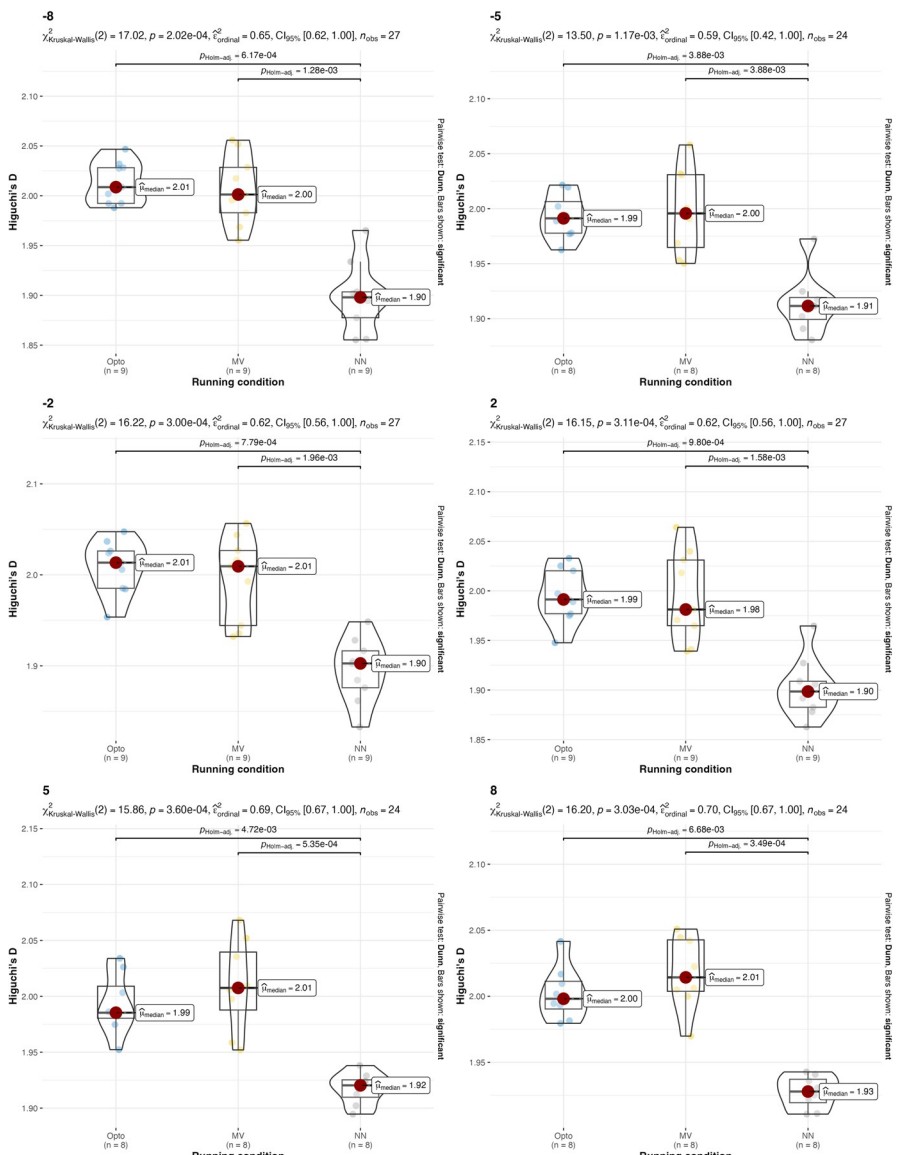

**Fig 9. Values and results of the statistical analysis for the running variability (HG-D) in different treadmill inclinations.** Values are reported for the three estimation strategies: the gold-standard optoelectronic device (OPTO), the IMU sensors attached to the runners' feet (MV), and the single IMU sensor attached to the runner's thorax used as input for the machine learning algorithm (NN).

running. In line with what is reported in [22], IMUs placed at the foot instep level can provide better estimations for the stride frequency than for the duty factor. Falbriard and colleagues [22] suggested that, to compute the duty factor with an instep IMU, the maximum value of the vertical acceleration should be used. However, based on the authors' experiences, using the acceleration norm has proven to be a more reliable approach (this method was also mentioned by Falbriard et al.). Previous attempts to estimate the duty factor based on the peak of vertical acceleration have led to many incorrect estimations. However, it is possible that this is why the present study found a significant difference in duty factor estimation between the instep foot sensors and the other two estimation strategies. In a recent and well-conducted research, Patoz et al. [30] compared the DF estimation with different machine learning models (one of the

models was a two-layer neural network) receiving the data from an IMU positioned at the runners' sacral level: the error in the estimation of DF reported was in line with what was found here (1.7±0.1%).

When it comes to running variability, the values reported in the literature suffer from a significant amount of variability around the average values. This means that the absolute DFA-α values frequently overlap between studies, which makes it challenging to determine a minimum worthwhile difference and the acceptable estimation error. Additionally, there is a lack of information in the literature regarding the potential use of IMU configurations for assessing running variability. For example, Norris et al. [14, 15] developed and then tested a new system based on an IMU positioned on the runner's tibia to estimate running variability in real-time. Recently, Hunter et al. [16] used a couple of IMUs placed on each runner's foot to monitor DFA-α during an outdoor marathon. No meaningful relationship between DFA-α and distance was found, somehow supporting the results reported by Norris et al. [14]. Unfortunately, in both these studies, the DFA-α estimations were not compared to a gold-standard estimation strategy (e.g., optoelectronic device or force platform). Meardon et al. [11], back in 2011, adopted a mono-axial accelerometer on the runners' tibia to monitor DFA-α during a prolonged indoor track run. However, Meardon et al. [11] did not report any data about the expected accuracy of the predictions, and indeed the value reported for DFA-α were surprisingly high at the beginning of the run (~1.19 for a group of healthy recreational runners). Mann et al. [13] adopted a gold-standard approach (i.e.: a motion capture system) and reported much smaller values. In absence of validation studies, it is difficult to establish whether the differences in DFA-α reported in the literature [11, 31, 32] are due to the different estimation strategies or the underlying locomotor control system.

The results suggest that both the IMU-based estimation strategies presented in this manuscript can be used, under certain conditions, as an alternative to gold-standard expensive instrumentations (e.g., opto-electronic devices and force platforms). Confirming what is already known in the literature [18], both IMU-based strategies have been found to be effective in the estimation of stride frequency (**Table 2** and **Figs 2** and **3**). Results were consistent across different running speeds and treadmill inclinations. In the estimation of the duty factor, the estimation strategy with the IMU on the feet was found to be not accurate across all the different running speeds and treadmill inclinations (**Table 2** and **Figs 4** and **5**). More precisely, the estimation strategy of the IMU placed on the thorax and the machine learning algorithm provided much better estimations of the duty factor in all the running conditions, except for the slowest running speed (80%PRS). In the estimation of the running variability from stride frequency evaluated with short 2-second time-window, IMU-based estimation strategies provided overall good accuracy for DFA-α. However, the method that makes use of the IMU on the runner's thorax and the machine learning algorithm provided significantly misleading estimations for DFA-α in downhill running conditions (-8 and -2) (**Table 2** and **Figs 6** and **7**). Good estimations of HG-D were returned by the method of the IMU attached at the runners' feet. Conversely, misleading estimations of HG-D were returned by the IMU placed on the running thorax and the machine learning model (**Table 2** and **Figs 8** and **9**). However, the results were sufficiently good only for the uphill conditions. Given that DFA-α and HG-D values did not fluctuate much across different speeds, IMU methodologies come with a warning: it might be better adopted in monitoring variations of DFA-α or HG-D over time, rather than to assess the absolute instantaneous variability value. In addition, the estimations errors [28] might be too large to detect the meaningful differences found with gold-standard instrumentation in uphill vs downhill treadmill running variability.

In terms of predicting DFA-α, utilizing a machine learning algorithm that incorporates features computed across a 120-second window can significantly enhance the accuracy of the

predictions (as shown in **Table 2**). This improvement becomes evident when comparing this estimation method with the gold standard optoelectronic device across various running conditions, as depicted in **Figs 6** and **7**. It is important to note that estimating DFA-α solely from a frequency computed from a rolling 2-second window may be inadequate, as this approach may overlook the long-term relationships present in the signal. To overcome this limitation, it is recommended to employ a longer time window, such as 120 seconds, to extract features that encompass more comprehensive information about the signal's long-term relationships. By doing so, the analysis becomes more complete and yields more accurate estimates of DFA-α. Thus, the model that receives the short 2-second window can be utilized to estimate stride frequency and duty factor, while the model that operates on the longer 120-second window should be employed specifically for estimating DFA-α. In summary, the two models, operating on different time windows, can be considered complementary in the analysis. The shorter window captures immediate frequency-related information, while the longer window enables the extraction of features that capture the long-term relationships carried by the signal, resulting in more accurate DFA-α estimates. Both models can leverage the signal coming from a single IMU positioned on the runner's thorax.

The study has certain limitations that need to be acknowledged. The initial sample size of 20 runners was randomly divided into two groups, resulting in a sample of 10 runners for evaluating the accuracy of the estimation strategies. A larger sample size would have provided a more robust understanding of trends and differences in the temporal variables under investigation. Moreover, the machine learning model presented in this study was not compared to alternative models employing different architectures. Future research could explore more advanced models, such as temporal convolutional networks, to potentially improve the accuracy of the estimations. It is important to note that the accuracy of the estimations is limited to the specific machine learning model used in controlled laboratory conditions and treadmill running. Further studies are warranted to assess the performance of the machine learning algorithm in different settings (e.g., outdoor environments) and various running conditions or surfaces.

The values of treadmill running kinematics and variability reported here can be used to support future research, meta-analyses, and reviews on the topic. An innovative aspect of this study consists in the DFA-α and HG-D values for different treadmill inclinations. Additional insights are provided here about the accuracy of different estimation strategies that make use of IMU technologies. Future research and investigations that do not make use of gold standard laboratory technologies should take into considerations the accuracy of the different estimation strategies before designing observational or interventional studies on running variability.

## Supporting information

**S1 File.**
(PDF)

## Acknowledgments

The authors are deeply grateful to ANTA and Nicholas Tam for awarding the PRISSIV project with the ANTA Sports Award. The authors are thankful to Nico Kaartinen (Kaasa Solution) and Terho Lahtinen (Movesense) for their assistance during data collection and processing.

## Author Contributions

**Conceptualization:** Andrea Zignoli, Laurent Mourot.

**Data curation:** Andrea Zignoli.

**Formal analysis:** Andrea Zignoli.

**Funding acquisition:** Andrea Zignoli, Laurent Mourot.

**Investigation:** Antoine Godin.

**Methodology:** Andrea Zignoli, Antoine Godin, Laurent Mourot.

**Project administration:** Andrea Zignoli.

**Software:** Andrea Zignoli.

**Supervision:** Laurent Mourot.

**Writing – original draft:** Andrea Zignoli, Laurent Mourot.

**Writing – review & editing:** Andrea Zignoli, Laurent Mourot.

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
