## [Decision Letter · Decision Letter 0]

3 Apr 2023

PONE-D-23-07146Indoor running spatio-temporal variability: normative values for different running speeds, inclinations, and three different estimation strategiesPLOS ONE

Dear Dr. Zignoli,

Thank you for submitting your manuscript to PLOS ONE. After careful consideration, we feel that it has merit but does not fully meet PLOS ONE’s publication criteria as it currently stands. Therefore, we invite you to submit a revised version of the manuscript that addresses the points raised during the review process.

We look forward to receiving your revised manuscript.

Kind regards,

Alessandro Mengarelli

Academic Editor

PLOS ONE

Journal Requirements:

Additional Editor Comments:

Both Reviewers highlighted some issues that need to be carefully addressed before the paper can be considered suitable for publication. I would suggest to carefully consider all the concerns. In particular, one Reviewer raised significant concerns regarding also methodological aspects.

Reviewers' comments:

Reviewer's Responses to Questions

**Comments to the Author**

1. Is the manuscript technically sound, and do the data support the conclusions?

Reviewer #1: Yes

Reviewer #2: No

2. Has the statistical analysis been performed appropriately and rigorously? 

Reviewer #1: Yes

Reviewer #2: No

3. Have the authors made all data underlying the findings in their manuscript fully available?

Reviewer #1: Yes

Reviewer #2: Yes

4. Is the manuscript presented in an intelligible fashion and written in standard English?

Reviewer #1: Yes

Reviewer #2: Yes

5. Review Comments to the Author

Reviewer #1: The study is interesting and worthy of investigation. I would congratulate the authors as this work was the most reproducible and transparent work that I have reviewed, as all the codes and analysis are in repositories. My major comment is regarding the results section, when there is a lack of data to understand the results without the necessity to check the supplementary material (see specific comment below).

Introduction, end of the first paragraph: Please, include one or two sentences describing both parameters and the differences between them.

Methods, “age 25 (6); BMI 20.76 (1.75)”: Please, include the units of both variables. Moreover, I suggest avoiding the abbreviation of males and females.

Statistical analysis: I recommend including the effect size analysis to improve the interpretation of the results.

Results: Even if all the results are published in one report, this is not entirely easy for readers. I think that the authors should add some figures in the results section to show the data obtained more visually so that the values can be viewed comfortably. Currently, as the results section is, it is known in what they exist and not significant differences, but without accessing said report you cannot see the values, which is not valid. The article must contain all the essential information by itself without having to check the supplementary material.

Results, “-8 vs +2 (p=0.00237) and +5 (p=0.00106) and +8 (p=0.00749), -5 vs +2 (p=0.0087) and +5 (p=0.00467) and +8 (p=0.02), -2 vs +2 (p=0.02) and +5 (p=0.01).”: I recommend including the DFA-α values in the parenthesis to understand the differences between inclinations or include a figure with the data.

Discussion, first paragraph: I recommend including the main results observed by the study.

Discussion: include at the end a paragraph about the limitations of the study.

Reviewer #2: Please see attached files. Please see attached files. Please see attached files. Please see attached files. Please see attached files. Please see attached files. Please see attached files. Please see attached files.

6. PLOS authors have the option to publish the peer review history of their article (what does this mean?). If published, this will include your full peer review and any attached files.

Reviewer #1: No

Reviewer #2: **Yes: **Marco Caruso

---

## [Author Response · Author response to Decision Letter 0]

4 Jun 2023

Please find the attached rebuttal letter.

---

## [Decision Letter · Decision Letter 1]

20 Jun 2023

Indoor running temporal variability for different running speeds, treadmill inclinations, and three different estimation strategies

PONE-D-23-07146R1

Dear Dr. Zignoli,

We’re pleased to inform you that your manuscript has been judged scientifically suitable for publication and will be formally accepted for publication once it meets all outstanding technical requirements.

Kind regards,

Alessandro Mengarelli

Academic Editor

PLOS ONE

Additional Editor Comments (optional):

Both Reviewers, who are expert in the field, were fully satisfied by the authors revision of the manuscript. The paper was significantly improved and it is now suitable for publication.

Reviewers' comments:

Reviewer's Responses to Questions

**Comments to the Author**

1. If the authors have adequately addressed your comments raised in a previous round of review and you feel that this manuscript is now acceptable for publication, you may indicate that here to bypass the “Comments to the Author” section, enter your conflict of interest statement in the “Confidential to Editor” section, and submit your "Accept" recommendation.

Reviewer #1: All comments have been addressed

Reviewer #2: All comments have been addressed

2. Is the manuscript technically sound, and do the data support the conclusions?

Reviewer #1: Yes

Reviewer #2: Yes

3. Has the statistical analysis been performed appropriately and rigorously? 

Reviewer #1: Yes

Reviewer #2: Yes

4. Have the authors made all data underlying the findings in their manuscript fully available?

Reviewer #1: Yes

Reviewer #2: Yes

5. Is the manuscript presented in an intelligible fashion and written in standard English?

Reviewer #1: Yes

Reviewer #2: Yes

6. Review Comments to the Author

Reviewer #1: Congratulations to the authors for the work done. All my previous comments were perfectly addressed.

Reviewer #2: I congratulate the authors for their work. The modified version of the manuscript, especially the results section, is now much clearer. I have no other points to raise, the paper is complete.

7. PLOS authors have the option to publish the peer review history of their article (what does this mean?). If published, this will include your full peer review and any attached files.

Reviewer #1: No

Reviewer #2: **Yes: **Marco Caruso

---

## [Editor Report · Acceptance letter]

12 Jul 2023

PONE-D-23-07146R1 

Indoor running temporal variability for different running speeds, treadmill inclinations, and three different estimation strategies 

Dear Dr. Zignoli:

I'm pleased to inform you that your manuscript has been deemed suitable for publication in PLOS ONE. Congratulations! Your manuscript is now with our production department. 

Kind regards, 

on behalf of

Dr. Alessandro Mengarelli 

Academic Editor

PLOS ONE